# Biological Synthesis, Characterization, and Therapeutic Potential of *S. commune*-Mediated Gold Nanoparticles

**DOI:** 10.3390/biom13121785

**Published:** 2023-12-13

**Authors:** Yaser E. Alqurashi, Sami G. Almalki, Ibrahim M. Ibrahim, Aisha O. Mohammed, Amal E. Abd El Hady, Mehnaz Kamal, Faria Fatima, Danish Iqbal

**Affiliations:** 1Department of Biology, College of Science Al-Zulfi, Majmaah University, Majmaah 11952, Saudi Arabia; ai.osman@mu.edu.sa (A.O.M.); a.elhady@mu.edu.sa (A.E.A.E.H.); 2Department of Medical Laboratory Sciences, College of Applied Medical Sciences, Majmaah University, Majmaah 11952, Saudi Arabia; sg.almalki@mu.edu.sa; 3Department of Pharmacology, Faculty of Medicine, King Abdulaziz University, Jeddah 21589, Saudi Arabia; imibrahim1@kau.edu.sa; 4Department of Pharmaceutical Chemistry, College of Pharmacy, Prince Sattam Bin Abdulaziz University, Al-Kharj 11942, Saudi Arabia; mailtomehnaz@gmail.com; 5Department of Agriculture, Integral Institute of Agriculture, Science and Technology, Integral University, Lucknow 226026, India; 6Department of Health Information Management, College of Applied Medical Sciences, Buraydah Private Colleges, Buraydah 51418, Saudi Arabia; danish.khan@bpc.edu.sa

**Keywords:** antifungal, cytotoxic, gold nanoparticles, *Schizophyllum commune*

## Abstract

Green-synthesized gold nanoparticles demonstrate several therapeutic benefits due to their safety, non-toxicity, accessibility, and ecological acceptance. In our study, gold nanoparticles (AuNPs) were created using an extracellular extract from the fungus *Schizophyllum commune* (*S. commune*). The reaction color was observed to be a reddish pink after a 24 h reaction, demonstrating the synthesis of the nanoparticles. The myco-produced nanoparticles were investigated using transmission electron microscopy (TEM), dynamic light scattering (DLS), and UV–visible spectroscopy. The TEM pictures depicted sphere-like shapes with sizes ranging from 60 and 120 nm, with an average diameter of 90 nm, which is in agreement with the DLS results. Furthermore, the efficiency of the AuNPs’ antifungal and cytotoxic properties, as well as their production of intracellular ROS, was evaluated. Our findings showed that the AuNPs have strong antifungal effects against *Trichoderma* sp. and *Aspergillus flavus* at increasing doses. Additionally, the AuNPs established a dose-dependent activity against human alveolar basal epithelial cells with adenocarcinoma (*A549*), demonstrating the potency of synthesized AuNPs as a cytotoxic agent. After 4 h of incubation with AuNPs, a significant increase in intracellular ROS was observed in cancer cells. Therefore, these metallic AuNPs produced by fungus (*S. commune*) can be used as an effective antifungal, anticancer, and non-toxic immunomodulatory delivery agent.

## 1. Introduction

Human health can be negatively affected by several infectious and metabolic diseases, including fungal infections, which are responsible for more than 1.7 million deaths. Nearly one billion people are infected each year, where cancer is a leading cause of death worldwide, accounting for nearly 10 million deaths in 2020 [1]. Although numerous therapeutic drugs have been created to address infectious and metabolic diseases, their practical implementation remains significantly constrained [2]. A breakthrough in biological research has begun with the production of nanomaterials and nanoparticles, which have potential uses in biomedicine, pharmaceuticals, biological sensors, skincare products, food technology, electronic devices, optoelectronic devices, dye degradation, the treatment of wastewater, etc. [3]. Nanoparticles (NPs) are extremely tiny materials, ranging in size from 1 to 100 nm. Based on their characteristics, shapes, or sizes, they can be divided into many classifications [4]. Fullerenes, metallic NPs, ceramic NPs, and polymeric NPs are some of the different categories of nanoparticles. Due to their large surface area and nanoscale size, NPs have distinct physical and chemical characteristics [5].

Gold nanoparticles possess exceptional properties including a regulated size, stability, and biocompatibility, and could be used to treat and diagnose cancer [6,7]. Previous reports have suggested that reducing the corresponding gold salts of nanoparticles is the simplest way to create them via chemical, physical, and biological processes. Physical and chemical processes include chemical reduction, microemulsion, cavitation, irradiation, electrochemical and microwave-assisted processes, the Turkevich method, laser ablation, and high-intensity ball milling, which have all been found to be costly, hazardous, and chemically damaging [8]. Presently, the realm of research is actively exploring biologically synthesized gold nanoparticles, which hold substantial promise for biomedical applications [9]. Bioactive compounds from natural products and nanoparticles synthesized from them have several therapeutic properties, including antimicrobial, anticancer, antioxidant, and neuroprotective characteristics [10,11,12,13,14,15,16,17]. The creation of these nanoparticles involves utilizing biological techniques, including plant extracts and microorganisms such as bacteria, yeasts, and fungi [5]. These novel methods provide an environmentally friendly, cost-effective, and sustainable alternative to traditional chemical synthesis approaches [18]. Gold nanoparticles find diverse applications in drug delivery systems, diagnostics, therapeutics, and imaging technologies, underscoring their versatility and potential in addressing modern biomedical challenges [19]. Numerous biological approaches have been employed to create nanoparticles in clean, non-toxic, safe, biocompatible, and ecologically acceptable ways (Appendix A) [20]. Metal ion accumulation by these organisms has been viewed as an economical, ecologically benign, and readily achievable occurrence [21,22,23]. It was observed that fast metal ion reduction produced stable NPs with a range of various shapes and sizes.

Fungi are an appealing choice for generating different NPs due to their excellent metallic tolerance, great efficacy at attaching to the cell wall, rapid up-scaling, capacity to produce enormous amounts of enzymes, and capability of accumulating metals through physico-chemical and biological processes [24]. Extracellular proteins with a variety of attributes are released in large quantities by fungi. All of the proteins and extracellular enzymes involved in metal salt reduction and capping in NPs are collectively referred to as the “secretome” and are released into the extracellular environment [25]. In order to biosynthesize gold nanoparticles, these fungal filtrates contain NADPH enzyme-dependent reductase that converts Au^3+^ to Au^0^ through the enzymatic process of metal reduction [26,27,28]. *Aspergillus terreus* has been used in the biological synthesis of gold nanoparticles (AuNPs), which led to the development of microbicidal characteristics that were specifically directed against *Escherichia coli*, a pathogenic Gram-negative bacterium [29]. Additionally, *Penicillium rugulosum* has been used to synthesize gold nanoparticles, and an endophytic fungus, *Fusarium acuminatum,* has been used to synthesize silver nanoparticles. These green-synthesized NPs offer a wide range of potential applications in combating human pathogenic organisms and cancer pathogenesis. There has been significant advancement in the application of AuNPs produced in different shapes, like spheres, nanorods, nanoshells, nanocages, etc., in cancer therapies [28,30,31].

*S. commune* belongs to the large and remarkable group of mushrooms, and it is known as the split gill mushroom. It is able to produce a variety of useful metabolites and is known to possess various medicinal properties, such as antimicrobial, anti-inflammatory, and antiparasitic characteristics [32]. Previously, we have reported mycologically synthesized silver and copper nanoparticles by utilizing *S. commune* fungal extracts, which can be used in both medical and non-medical applications due to their potential antibacterial and antibiofilm properties against multidrug-resistant microorganisms [33]. However, there was no information available on the synthesis of AuNPs by utilizing *S. commune*.

Therefore, our group planned to focus on novel therapeutic approaches by applying green synthesis of nanomaterials with the help of fungus extracellular materials and elucidate their antifungal and cytotoxic potentials. Thus, the goal of the current work is to create AuNPs from extracellular matrix of *S. commune*. These mycologically synthesized gold nanoparticles have undergone evaluation for their potential antifungal and cytotoxic activities as well as intracellular ROS production in *A549* cell lines.

## 2. Materials and Methods

### 2.1. Myco-Synthesis of AuNPs from S. commune

The fungus was cultivated in MGYP broth (Hi-media), which consisted of malt extract, glucose, yeast extract, and peptone (0.5%, 1%, 0.3%, and 0.5%). The culture was maintained at 28 °C for an additional 72 h to allow for full growth before being harvested by straining through a polypropylene sieve. Using Whatman filter paper, 20 gm of mycelia was then collected, rinsed with sterile water, and transferred to 150 mL of deionized water. To secrete extracellular enzymes, the mycelial biomass was additionally stirred at 150 rpm for 72 h at pH 7.2 and 28 °C [27,28]. When 1 mm hydrogen tetrachloroaurate (HAuCl_4_) was added, the process continued with the production of AuNPs. After drying the sample, we obtained gold nanoparticles. and thereafter we made certain dilutions. Using “UVWinlab” software (version 1.05), the information was further examined. Salt-free supernatant was utilized in a Beckman DU-20 UV–Vis spectrometer, SpectraLab Scientific Inc., Markham, ON, Canada, as a control.

### 2.2. Characterization of Nanoparticles

#### 2.2.1. Ultraviolet–Visible Spectroscopy

The emergence of reduced gold nanoparticles in colloidal suspension has been observed using the Beckman DU-20 spectrophotometer. For AuNPs, the absorbance spectra were measured in the range of 400–650 nm [34]. Using the “UVWinlab” tool (Version 1.05), the outcomes were further explored and documented. A Beckman DU-20 UV–Vis spectrometer was used to measure distilled water as the standard, and the salt-free supernatant served as the negative control in the experiment.

#### 2.2.2. Differential Light Scattering (DLS)

Before conducting the size measurements, gold nanoparticles (AuNPs) were prepared in double-distilled sterile water (dH_2_O) using a bath sonicator (ULTRAsonik 57 X, 50/60 Hz, NEY DENTAL INC, Bloomfield, MI, USA). The viscosity of dH_2_O at 25 °C was determined through viscosity measurements (Viscometer-SV-10; A&D Instruments Ltd., Abingdon, UK), and these recorded values were used for all dynamic light scattering (DLS) size calculations. The viscosity of dH_2_O at 25 °C was measured to be 0.887 centipoise. The average diameter, peak diameter distribution, and polydispersity index (PdI) were determined using a Malvern Zeta Sizer-Nano ZSTM (Malvern Instruments, Malvern, UK) with Dispersion Technology Software v.5.1 (Malvern Instruments Ltd., Malvern, United Kingdom. The scale range for PdI was set from 0 to 1. Prior to each measurement, the samples for DLS were equilibrated at 25 °C for 2 min. The powder of the sample was further diluted to a concentration of 0.5% (*w*/*v*) in deionized water and sonicated for 1 min prior to estimation. The sample was placed in a DTS0112 low-volume dispensable measuring cuvette of 1.5 m. The refractive index (RI) of AuNP.dH_2_O was determined to be 1.4303 [35].

#### 2.2.3. Transmission Electron Microscopy Analysis (TEM)

Following the transfer of the synthesized AuNPs onto a gold-coated negative grid, the solvent was allowed to evaporate. The TEM examination was performed using a Perkin-Elmer model (JEM-1000; JEOL (UK) Ltd., Welwyn Garden City, UK) operating at an accelerating voltage of 1000 kV.

### 2.3. Antifungal Efficacy

The antifungal potential of biologically synthesized AuNPs was assessed against *Trichoderma* (NCIM, Accession No: 1458) and *A. flavus* (NCIM, Accession No: 1316), obtained from the National Center of Cell Science (NCCS). Inoculums containing 10^7^ spores/mL were prepared in sterilized phosphate-buffered solution (PBS) with a pH of 7.0 and recorded using a cell counter. Each fungal solution (1 mL) was evenly spread on PDA plates. Wells with a diameter of 0.5 cm were created using a sterile cork borer and filled aseptically with control (fungal extract) at 50 μg/mL, gold precursor [HAuCl_4_] (1mM at 50 μg/mL) and AuNPs at various concentrations (1.5 µg/mL, 5.7 µg/mL, 8.6 µg/mL, and 11.4 µg/mL), antibiotic (fluconazole) at 50 μg/mL (Hi-media), AuNPs + antibiotic (fluconazole) at 1.5 + 50 μg/mL. The plates were incubated at 28 ± 4 °C for 7 days, and the average inhibition zone was determined for each case. A salt-free filtrate was used as a negative control [36].

### 2.4. Preparation of Fungal Cells for SEM Analysis

The morphology of fungal cells and the cellular alterations induced by AuNPs in *A. flavus* were examined using scanning electron microscopy (FESEM Model No. GEMINISEM 300). The plates were subjected to a laser for a short period of time to observe the morphological modifications in fungal mycelia and spores caused by AuNPs at 8.6 µg/mL. As a control, untreated samples maintained in nutrient broth were used. These fungus cells were washed and then again resuspended in PBS buffer. After that, samples were placed on membrane filters and fixed for 4 h in 2% (*v*/*v*) glutaraldehyde. Following that, samples were washed repeatedly in phosphate-buffered saline (PBS) and allowed to fix for 1 h in 1% (*w*/*v*) osmium tetra oxide. Further, the solvent was removed by using ethanol series at various concentrations, including 25%, 35%, 55%, 75%, 85%, and 100%, before being gold-plated [37]. The semi-quantitative chemical composition of each sample was determined using a SEM device connected to an energy dispersive X-ray microscope–EDX (OCTANE ELECT PLUS), AMETEK Instruments India Pvt Ltd., Bangalore, India.

### 2.5. Cytotoxicity Assay

#### 2.5.1. Maintenance of Cell Lines

The Central Drug Research Institute’s Animal Tissue Culture laboratory (CDRI-ATCL) maintained a cell line of lung cancer epithelial cells termed *A549* (86012804-CDNA-20UL). Using conventional cell culture techniques, cells were kept alive in Dulbecco’s modified Eagle’s medium (DMEM) along with 1% antibiotic–antimycotic solution and 10% fetal calf serum (FCS) [38].

#### 2.5.2. MTT Assay

The cellular viability was evaluated using the 3-(4,5-dimethylthiazol-2-yl)-2,5-diphenyltetrazolium bromide (MTT) transformation test. Initially, cells were plated at a density of 1 × 10^4^ cells/mL on 96-well culture plates and incubated at 37 °C for 24 h in an incubator, allowing them to grow up to 80% confluence. After the incubation period, the medium was removed, and the cancer cells were treated with biosynthesized AuNPs at increasing concentrations of 5–25 μg/mL in triplicate; non-treated cells were used as a control and incubated for 24 h. Each well received MTT dye, and the plate was allowed to incubate at 37 °C for 24 h. Using a PowerWave XS “BIOTEK, USA” spectrophotometer, the absorbance of insoluble formazan salts was observed at 550 nm. The data used to form a dose–response graph were used to compute the dose of these metallic nanoparticles required to kill half of the cell population (IC_50_) [39].
Cell viability (%) = [Mean OD/Control OD] × 100

To perform the morphological assay, *A549* cells were exposed to nanoparticles (NPs) at different concentrations. Following exposure, the cells were fixed using a solution of ethanol and acetic acid in a ratio of 3:1 (*v*/*v*). Subsequently, a cover slip was delicately mounted on a glass slide to examine the morphological alterations.

### 2.6. ROS Estimation

Intracellular oxidative stress was assessed using fluorescent dye (DCFH-DA: 2′,7′-dichlorofluorescin di-acetate) that is known as a fluorescent indicator for the study of internally generated hydroperoxides [40,41]. The experiment was performed according to the methods described by Goswami et al. [42] with minor modifications. Briefly, a specific concentration (1000 cells/well) of *A549* cell lines was seeded in 96-well black-bottomed plates and allowed to adhere for 24 h before being exposed to AuNPs. Subsequently, *A549* cells were plated and dispersed in triplicate for the purpose of quantifying ROS. We used a negative control containing 20 µM DCFH-DA solution in PBS, while the positive control had 20 µM DCFH-DA solution in PBS with 1 µM hydrogen peroxide (H_2_O_2_). In the test samples, we incubated the positive control mixture with AuNPs at two concentrations (15 µg/mL and 25 µg/mL) for 2–6 h. A “BIOTEK-FLX800-USA” fluorometer with emission at 520 nm was used to measure the fluorescence intensity of cells to track the pace of intracellular oxidative stress (by 485 nm excitation).

### 2.7. Statistical Analysis

Three replicates were conducted for each study and the results were represented as the mean ± standard deviation of three individual experiments where * *p* < 0.05; ** *p* < 0.01; *** *p* < 0.001 represent significant difference from control group. To analyze the data with normally distributed values and homogeneous variance, one-way analysis of variance (ANOVA) with a *t*-test was performed.

## 3. Results

### 3.1. Preparation of AuNPs and Characterization of AuNPs

#### 3.1.1. Surface Plasmon Resonance

Gold nanoparticles (AuNPs) were mycologically synthesized using *S. commune* fungi. The fungal filtrate was utilized as a reducing agent in 1 mM gold-chloride tetrahydrate solutions. As the biosynthesis progresses, the color shifts from light yellow to red in about 24 h. A control solution of 1 mM HAuCl_4_ without fungal extract did not demonstrate any color change when subjected to similar conditions (Figure 1A).

#### 3.1.2. Spectrophotometric Analysis

In the 400–650 nm spectrum, the fungal extract shows no sign of absorbance. However, fungal extracts treated with HAuCl_4_ salt exhibit a prominent absorption peak at 545 nm, strongly suggesting the successful synthesis of AuNPs (Figure 1B). Similar results have been seen in earlier reports, where it has been demonstrated that the absorbance of gold NPs was in the range of 540–570 nm [43,44,45].

#### 3.1.3. Differential Light Scattering of AuNPs

The results presented in Figure 2 show the scattering intensity as a function of the logarithm of the particle diameters. In both cases, bimodal distributions were observed, with the peaks of the larger diameter exhibiting most of the intensity. The average particles size is approximately 90 nm, having a spherical morphology. We observed a peak from 40 nm to 250 nm in the DLS parameter, and this distribution of size may be due to the aggregation of nanoparticles. However, we have noticed the highest percentage peak intensity for particle sizes between 90 nm and 105 nm. The nanoparticles synthesized by this fungal mode were found to be homogeneous and show various sizes. Thus, the result clearly shows that most NPs were spherical in shape and have different size ranges. These synthesized AuNPs were polydisperse in nature as their PDI value was 0.3.

#### 3.1.4. Transmission Electron Microscopic (TEM) Analyses

The specimen photos taken from the drop-coated surface of the AuNPs uniformly dispersed over the grid further supported the observation made by TEM micrographs that they have a circular form. The AuNPs ranged in size from 60 nm to 120 nm, while the average size of nanoparticles is around 90 nm (Figure 3). Similar results were seen in DLS data of the diameter of each individual particle, helping the analysis of particle size.

### 3.2. Antifungal Activity of AuNPs

In this study, the inhibitory activity of AuNPs on colony development was studied under in vitro conditions.

#### 3.2.1. Agar Well Assay

The findings showed that the development of *A. flavus* and *Trichoderma* sp. was suppressed, creating an inhibition zone by AuNPs under in vitro conditions, demonstrating significant (*p* < 0.05) antifungal effect on *A. flavus* and *Trichoderma* sp. as well as inhibitory actions on mycelial hyphae and spore germination (P). According to earlier studies, *S. commune* extract has very low antifungal activity against pathogenic fungi [46,47]. In this instance, the values for the zone of inhibition (ZOI) of the fungal filtrate assumed as control are taken to be 0 cm. Similarly, the gold precursor did not exhibit any antifungal activity. Furthermore, it was found that as the concentration of AuNPs increased, there was a decrease in the spore-forming ability of the organisms. At a concentration of 11.4 µg/mL, *Trichoderma* sp. exhibited an inhibition zone with a diameter of 2.3 cm, while *A. flavus* displayed an inhibition zone measuring 2.7 cm. However, when these AuNPs are conjugated with antibiotic fluconazole, the ZOI is increased by 2.5 cm and 2.9 cm in *Trichoderma* sp. and *A. flavus*, respectively (Figure 4) (Appendix A). Fluconazole is a member of the triazole family and one of the most popular antifungal medications. It is a prominent medicine that most researchers use in their studies [48]. However, secondary metabolites present in the fungal extract of *S. commune* at a very higher concentration might inhibit the growth of the pathogenic fungi [46,49].

#### 3.2.2. Scanning Electron Microscopy (SEM) and Energy Dispersive Spectra (EDX) of Fungi Treated with AuNPs

SEM examination was carried out to observe the inhibitory effect of AuNPs (11.4 µg/mL) on the spores of *A. flavus*. The hyphae seemed distorted, smaller, less rigid, and lacking in structure. Additionally, spore wall disintegration and damage were seen following treatment with AuNPs in varied degrees. The cytoplasm of the spore wall displayed significant pitting, ripping, and piercing in addition to the visible uneven and rough cell walls and widespread blebbing (Figure 5). Clusters of the spore’s outermost layer also began to form when the outer wall’s integrity was visibly disturbed [50].

Mycelial abnormalities in the EDX spectra demonstrated the creation of AuNPs, in contrast to the microorganism utilized in this investigation, as they contained a peak consistent with the formation of pure gold (Figure 6). The EDX spectrum also exhibits peaks for carbon, nitrogen, sodium, and oxygen, indicating the existence of proteins and several fungal remnants within in the interstitial spaces of the fungus.

### 3.3. In Vitro Cytotoxicity Assay

In order to create a xenograft lung cancer model, particularly for non-small cell lung cancer (NSCLC), the *A549* cell line is a well-known and commonly utilized human cell line. For the cytotoxicity assay, we opted for *A549* cell lines as they are very prominent and have been widely used in cancer research [51]. Thus, by using the cell viability MTT test, we further evaluated the cytotoxicity of AuNPs on *A549* cells. Up to concentrations of 25 µg/mL, AuNPs exhibited maximum cytotoxicity, whereas the viability of cells exhibited a notable decrease when the doses reached 15 µg/mL, 20 µg/mL, and 25 µg/mL (Figure 7).

In our investigation, *A549* cells treated with 3-(4,5-dimethylthiazol-2-yl)-2,5-diphenyl tetrazolium bromide revealed morphological changes caused by AuNPs, as shown in Figure 8. MTT was used to dye *A549* cells, where the treatment caused most of the cells to shrink and separate from the culture dish’s substratum, while the control cells maintained their original appearance. To further confirm the results obtained from treating malignant cell lines with metallic nanoparticles, the effects of 5 µg/mL and 25 µg/mL AuNPs on cell morphology were examined. The cells exhibited significant alterations, including a noticeable reduction in cell membranes, indicating severe cellular damage and cell death, along with the presence of other cellular debris. AuNPs were shown to be toxic to cells at a concentration of 5 µg/mL and more toxic at 25 µg/mL, where the allegation that malignant cells are killed by metallic nanoparticles has been supported by the changes in the cell shape and a reduction in cell quantity. The synthesized gold nanoparticles appeared to be more toxic towards cancerous cells when compared to control cells (Figure 8).

### 3.4. ROS Activity

The intracellular level of reactive oxygen species of control and AuNP-treated (15 and 25 μg/mL) lung carcinoma *A549* cells is shown in Figure 9. We have observed a significant (*p* < 0.001) rise of around 40% and 60% in ROS content which was observed in cells treated with 15 and 25 μg/mL of AuNPs, respectively. It indicates the instigation of oxidative stress by the biosynthesized AuNPs that was the reason for the increased number of apoptotic cells observed in DCFDA staining (Figure 10).

The production of ROS is a sign of oxidative stress, where biological components suffer oxidative damage, which finally results in cell death. One of the main causes of AuNP toxicity is oxidative stress over cancerous cells, which can trigger apoptosis in response to a number of signals [39,52]. DCFH-DA has been utilized in our research to determine ROS generation. Fluorescence pictures of *A549* cells were obtained after 4 h with AuNP concentrations of 0 µg/mL (control) and 15 µg/mL. Unlike the AuNP-treated cells, the control sample did not exhibit any green fluorescence, indicating that H_2_O_2_ was not formed (Figure 10).

## 4. Discussion

### 4.1. Preparation of AuNPs and Characterization of AuNPs by Surface Plasmon Resonance, Spectrophotometric Analysis, Differential Light Scattering, and Transmission Electron Microscopic (TEM) Analysis

Gold nanoparticles (AuNPs) were mycologically produced using *S. commune* fungal extract. The appearance of a red color in the solution provided convincing evidence that AuNPs were being synthesized inside the reaction mixture [53,54,55]. The color shift is caused by the surface plasmon resonance of the metallic gold nanoparticles. The reduction of HauCl_4_ salts to AuNPs is supported by a previous report on *Yarrowia lipolytica*, where the contribution of redox mediators for AuNP production from a fungal source was observed. Melanin was secreted by that fungus, and intriguingly, it seems to convert Au^3+^ to AuNPs [56,57]. The enzymes nicotinamide adenine dinucleotide (NADH)/nicotinamide adenine dinucleotide phosphate (NADPH) oxidoreductase, which can be found on the outermost layer of the cell or within the cytoplasm, catalyze the reduction of Au ions. In *R. oryzae*, X-ray photoelectron spectroscopy revealed that the concentrations of Au^+^ and Au^0^ changed as the biosynthetic process progressed, indicating that Au^3+^ ions were first reduced to Au^+^ and further to Au^0^ [57]. The absorption band’s intensity is increased with time and was maximum after 24 h. Beyond this point, no more fluctuations in the spectrum were observed, suggesting that the gold salt precursors had been fully consumed [58]. In DLS, we observed a peak from 40 nm to 250 nm, and this distribution of size may be due to the aggregation of nanoparticles. The primary cause of the size change is the formation of the corresponding nanostructures by the oxidation of metallic salts in the presence of protein [59]. This approach enables the assessment of random fluctuations in the light diffused from a colloidal suspension, providing a means to measure the particle size. The size variability is caused by the metal reduction to their corresponding NPs by the influence of reductase enzyme. This method allows for the measurement of the random variations in light intensity dispersed from a supernatant [39,60]. Similarly, in TEM micrographs, it was observed that the material produced using this biological mode was more homogeneous and had a number of particles of different sizes. This outcome demonstrates the varied size range and generally spherical shape of the particles. Similar results were observed in previous studies, which supports our findings [61].

### 4.2. Antifungal Activity of AuNPs by Agar Well Assay and Their Scanning Electron Microscopy (SEM) and Energy Dispersive Spectra (EDX)

The biological synthesis of nanoparticles offers a promising and advantageous alternative to traditional chemical methods due to its eco-friendliness, cost-effectiveness, milder reaction conditions, specificity, biocompatibility, and reduced generation of hazardous byproducts. The biologically produced nanoparticles were found to be polydisperse in nature. The fundamental properties, such as electronic, optical, magnetic, and catalytic properties, are controlled by the size and shape of the nanoparticle. The level of uniformity or dispersity in biologically produced nanoparticles can vary based on the specific biological method, the involved organisms or biological materials, and the process conditions [62,63]. A few studies also reported that biologically synthesized AuNPs might be a suitable alternative in comparison to chemically synthesized AuNPs for coating antimicrobial medicines in the pharmaceutical industry. As they are less toxic and less destructive to probiotics present in the human gut in the form of probiotic bacteria, the outer coatings of nanoparticles on the drugs may be more effective for destroying pathogenic bacteria and safe for humans [64]. Gold precursors, such as chloroauric acid, have limited antifungal effectiveness due to their molecular form, hindering direct interaction with fungal cells unlike nanoparticles. Unlike nanoparticles, gold precursors have smaller surface areas, necessary for effective contact with fungal cells. Moreover, their larger size and chemical structure limit penetration through fungal cell walls, reducing direct interaction and leading to their insufficient antifungal action [65,66].

Mycelial abnormalities in EDX spectra demonstrated the creation of AuNPs, with peaks for C, N, Na, and O_2_ indicating the existence of proteins and several fungal remnants present within in the interstitial spaces of the fungus. As stated in the procedure, the high carbon content of the growth media is a requirement for the fungus to carry out its metabolic activities.

### 4.3. In Vitro Cytotoxicity Assay

In our investigation, *A549* cells treated with 3-(4,5-dimethylthiazol-2-yl)-2,5-diphenyl tetrazolium bromide revealed morphological changes caused by AuNPs. Previous reports observed that AuNP-treated cells showed the existence of MTT-stained cells that had apoptotic aggregates, blebbing membrane, and constricted nuclei [67]. Moreover, husk-like zinc oxide nanoparticles showed promising significance for chemotherapy in MTT assay, the investigation of reactive oxygen species (ROS) release, and condensation of chromatin studies towards the human epidermoid carcinoma (HEC) *A431* cells [68]. According to several studies, *S. commune* extract has very low cytotoxic activity against cancerous cell lines. The cytotoxicity investigation indicated that 1 mg/mL of *S. commune* extract exhibit only 37% inhibitory effect on the survival of DF-1 cell lines, and it was concluded that the extract does not have significant cytotoxicity because there was no 50% cell death observed even at 100 mg/mL [69]. Similarly, MTT assays were also carried out on Chinese hamster ovary (CHO) cell lines that showed negligible toxicity of *S. commune* fungal extract [70].

The biologically produced gold nanoparticles were found to be significantly toxic to cancerous cells (MDA-MB-231) with an IC_50_ value of 43.09 ± 1.6 µg/mL. However, even at a higher concentration of 150 µg/mL, they exhibited minimum toxicity on human embryonic kidney cells. Morphological data also showed similar activity where nanoparticles caused apoptosis to kill cancer cells but had little or no negative impact on healthy cells, which could be a promising tool in a range of biomedical applications [71]. B16F10 (non-cancerous) cells exhibited greater resistance towards AuNPs than HeLa cells (cancerous cells) in cytotoxicity experiments. Decreased systemic cytotoxicity to non-cancerous cells allowed the 20 nm AuNPs to exhibit improved therapeutic efficacy. As a result, 20 nm AuNPs may be thought of as a great substitute for human cervix carcinoma treatment since they may be safely delivered into the bloodstream while having few adverse effects on non-cancerous cells. It has been summarized that AuNPs of various sizes, therefore, displayed varying degrees of cytotoxicity in various cell types [72]. Gold nanoparticles effectively reduced the proliferation of breast cancer cells throughout time and at different doses. El-Sayed et al. revealed that there was no cytotoxic effect on healthy cells (MCF-10A) [73]. Similar to this, the AuNPs absorb 6 times more in cancer cells than in healthy human cells. More crucially, they were able to demonstrate that cancer cells are more likely to be killed than normal cells after four minutes [74].

Based on the dose-dependent suppression of *A549* cells’ proliferation, the MTT experiment demonstrated that AuNPs are hazardous. In *A549* cells, treatment with AuNPs upregulates the expression of antiapoptotic proteins while activating caspase expression [52]. In a few studies, it was clearly mentioned that the AuNPs exhibit selectivity in the toxicity effect against cell lines. The biosynthesized AuNPs were present in the cytosol, and their selectivity toward cancer cells can be optimized for their potential use in biomedical research on cell biology, cancer therapy, and targeted drug delivery. Similar research was performed that showed the effect of AuNPs on human fibroblast cell line CIRC-HLF [75], cervical cancer cells (HeLa) [76,77], and a human colorectal carcinoma cell line (HCT-116) [78] showing toxicity in a dose-dependent manner. Various reports suggested that cytotoxicity also depends on the type of cells used. The 33 nm citrate-capped AuNPs were found to be non-toxic to baby hamster kidney and human hepatocellular liver carcinoma cells but toxic to a human carcinoma lung cell line [79]. Therefore, these studies provide clear evidence that metallic nanoparticles induce cell death, supporting our findings regarding the efficacy of myco-synthesized AuNPs against *A549* cells. This further emphasizes their potential as a valuable tool in the arsenal for cancer treatment.

### 4.4. ROS Activity

The production of ROS is a sign of oxidative stress, where biological components suffer oxidative damage, ultimately resulting in cell death. One of the main causes of AuNPs’ toxicity is oxidative stress over cancerous cells which can trigger apoptosis in response to a number of signals [39]. In *A549* cells treated with AuNPs, the greatest intensity of green fluorescence was seen at 4 h. Thus, cells exposed to AuNPs produced ROS that have the potential to cause cellular disruption and ultimately result in cell death [80]. These findings concur with the published studies that show DNA adducts were created as a result of the treatment with AuNPs, increasing the levels of intracellular ROS and eliminating antioxidants such as reduced glutathione or antioxidant enzymes like glutathione peroxidase and superoxide dismutase [81]. It has been suggested that intracellular ROS is a key indication of the toxic effect of NPs for cancerous cells. Recent research has shown that AuNP-mediated ROS production in various cell types led to cell death [82].

From previous studies, biological and chemically synthesized AuNPs were compared, and it was found that chemically produced nanoparticles (citrate-capped particles) were only able to kill about 20–25% of cancerous cells even at a high concentration of 400 µg/mL, while the IC_50_ value for biologically synthesized AuNPs was found to be around 200 µg/mL. The greater cytotoxic efficiency of biologically synthesized AuNPs compared to chemically synthesized AuNPs was demonstrated by colony suppression tests, intracellular ROS quantification, deregulation of mitochondrial membrane potential, and cell cycle arrest [83]. Due to stabilization by biological metabolites, AuNPs contribute significant cytotoxic activity. A few studies have suggested that biologically manufactured AuNPs would be a better choice for coating antimicrobial drugs in the pharmaceutical business than chemically synthesized AuNPs [84]. The exterior coatings of nanoparticles on the medications may be more effective at killing pathogenic microbes and safe for humans since they are less toxic and harmful to probiotics existing in the human gut. These findings point to the possible application of AuNPs as an antifungal agent in the biomedical field. Furthermore, since these AuNPs are readily conjugated with drugs, they can also function as an excellent drug in biomedical applications [85,86].

Considering the reduced cytotoxicity of biologically synthesized gold nanoparticles (AuNPs) compared to chemically prepared ones, the size of the particles plays a significant role. Biologically synthesized AuNPs often exhibit a more controlled and uniform size distribution compared to their chemically synthesized counterparts [9]. The controlled size of biologically synthesized AuNPs is advantageous in mitigating cytotoxic effects. Smaller nanoparticles typically have larger surface-area-to-volume ratios, and their interactions with cells are influenced by factors such as cellular uptake, internalization, and bioavailability. Biologically synthesized AuNPs, often in the nanoscale range, are more efficiently internalized by cells due to their optimized size, promoting cellular compatibility [87]. In contrast, chemically prepared AuNPs may exhibit a broader size distribution, including larger particles that could induce increased cytotoxicity through mechanisms such as enhanced oxidative stress and inflammation [88].

Furthermore, the surface coating of biomolecules on biologically synthesized AuNPs not only enhances biocompatibility but also contributes to the overall stability of the nanoparticles. This stability is crucial for maintaining consistent size characteristics over time and under different environmental conditions [18]. In contrast, chemically prepared AuNPs may be more prone to agglomeration, leading to variations in size and potentially exacerbating cytotoxic effects. Thus, the controlled and often smaller size, along with the protective biomolecular coating, makes biologically synthesized AuNPs less cytotoxic to cells compared to chemically prepared counterparts [89]. These factors collectively contribute to the safer interaction of biologically synthesized AuNPs with biological systems, emphasizing their potential for applications in medicine and biotechnology.

## 5. Conclusions

In conclusion, *S. commune* fungus was effectively used to produce a simple, reliable, and ecologically friendly method of biosynthesizing AuNPs. UV–visible spectroscopy was employed to evaluate the intensity of the pink color at various wavelengths between 400 and 600 nm to determine the highest surface plasmon resonance (SPR). The greatest peak, as indicated by the UV–visible chart, was found at a wavenumber of 545 nm, corresponding to the SPR of AuNPs. DLS and TEM analysis indicated that mycologically synthesized AuNPs were spherical in shape within the size range of 60 nm to 120 nm. Additionally, as antifungal agents, these modified AuNPs were more effective against pathogenic fungus. After treatment with AuNPs, several mycelial abnormities were found during SEM analysis. Moreover, the AuNPs were also found to be cytotoxic against the *A549* cell line (adenocarcinoma alveolar basal epithelial cells). The fungal strain utilized in this work is therefore likely to have several benefits, including its efficiency in producing AuNPs, which can act as powerful fungicides and be used as a cytotoxic agent, leading to the production of intracellular ROS that can be helpful to cure cancer. Therefore, this method of AuNP production development leads to disease therapies that are less expensive, safe, biocompatible, and result in reduced generation of hazardous byproducts.

## Figures and Tables

**Figure 1 biomolecules-13-01785-f001:**
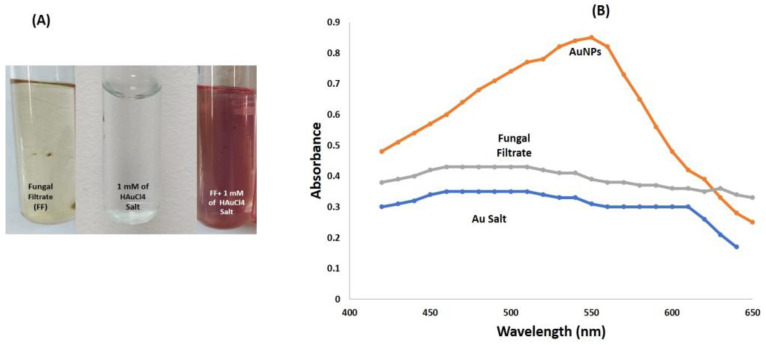
(**A**) Glass tubes showing gold solution with fungal filtrate, 1 mM solution of HAuCl_4_, and fungal mediated AuNPs (**B**) UV–Vis absorption spectra of AuNPs, HAuCl_4_, and fungal filtrate.

**Figure 2 biomolecules-13-01785-f002:**
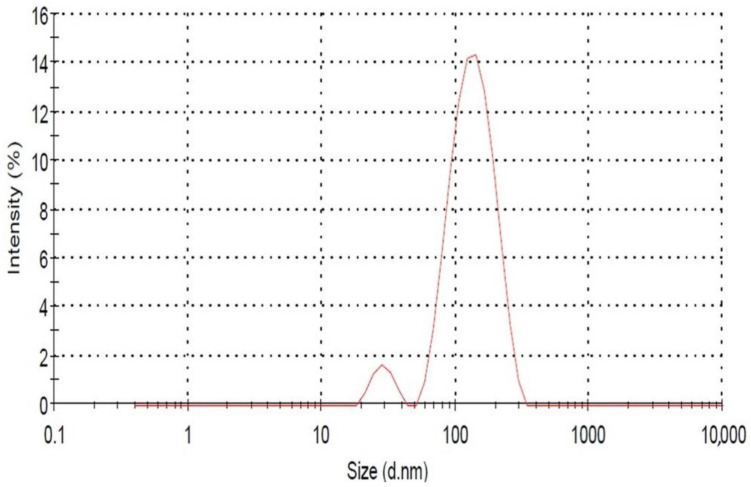
Distribution of gold nanoparticles analyzed through dynamic light scattering (DLS) data.

**Figure 3 biomolecules-13-01785-f003:**
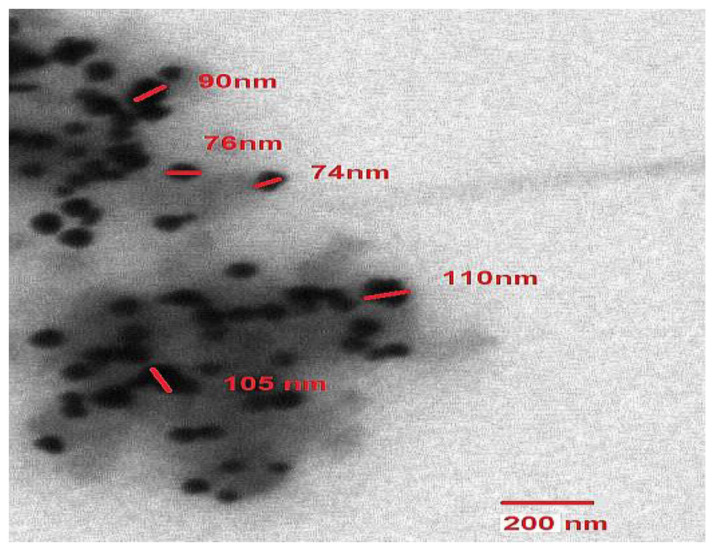
TEM images of spherical gold nanoparticles ranging from 60 nm to 120 nm produced by fungal extract.

**Figure 4 biomolecules-13-01785-f004:**
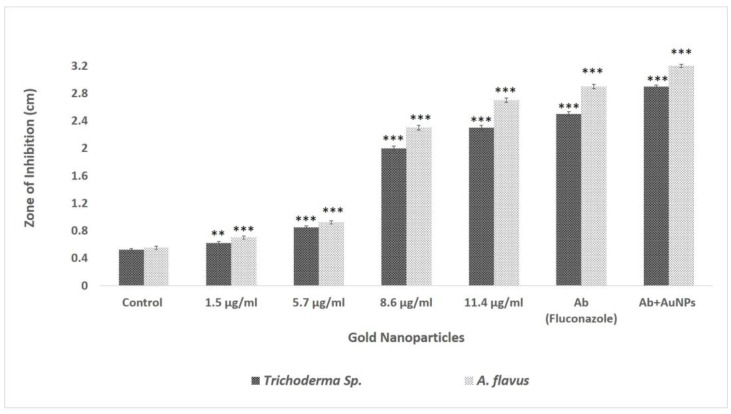
Antifungal activity of AuNPs on *A. flavus* and *Trichoderma* sp. Data represent the mean ± standard deviation of three individual experiments. ** and *** represent significant difference from control group (** *p* < 0.01; *** *p* < 0.001). The *X*-axis showing AuNP concentration does not only have AuNPs but also shows control (fungal extract), antibiotic (Ab), and combination (Ab + AuNPs).

**Figure 5 biomolecules-13-01785-f005:**
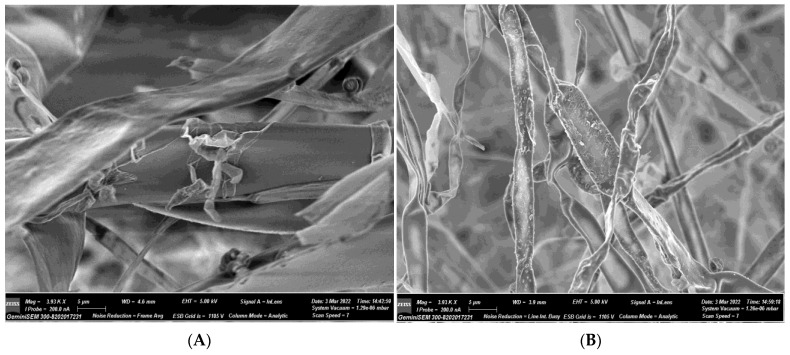
Scanning electron micrographs of *A. flavus* with 11.4 µg/mL of AuNPs. (**A**) Mycelia of *A. flavus* (Control: Working distance (4.6 nm)). (**B**) Treated: Working distance (3.9 nm) at 5 µm. (**C**) Spore of *A. flavus* (Control: Working distance (4.6 nm)). (**D**) Treated: Working distance (4.7 nm) at 1 µm. Magnification: 12.85 KX; Signal A = InLens; I Probe = 200.0 nA; ESB Grid = 1105 V; Column Analytic; GeminiSEM 300-8202017231 was used.

**Figure 6 biomolecules-13-01785-f006:**
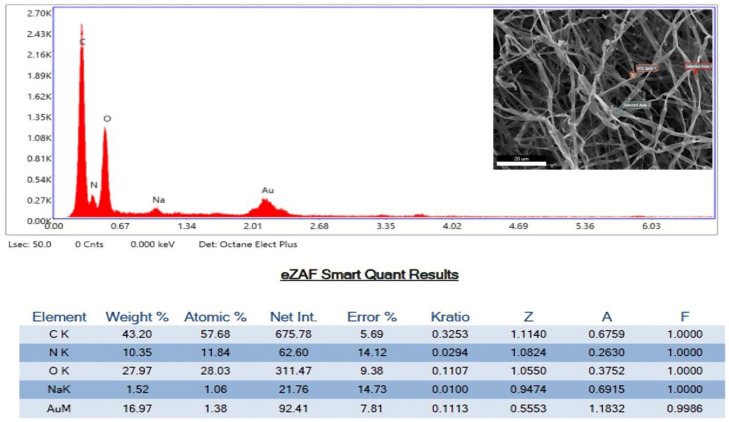
SEM images with EDX analysis of gold nanoparticles synthesized using fungal extract of *S. commune*. The graph represents the elemental analysis of AuNPs within the fungal mycelia while the inset picture contains the position where the AuNPs affected the fungal mycelia.

**Figure 7 biomolecules-13-01785-f007:**
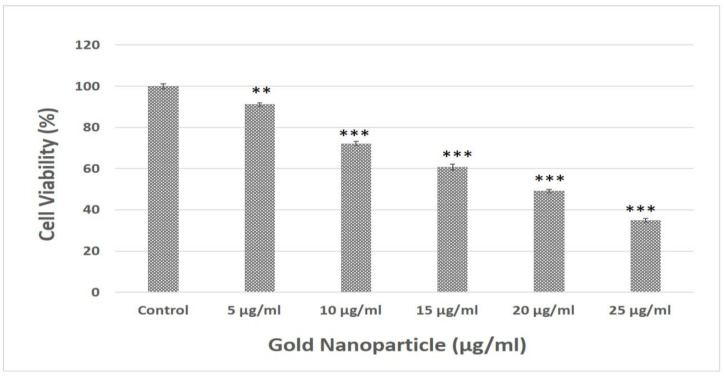
Effect of gold nanoparticle on cell viability of *A549* cancerous cell lines. Data represent the mean ± standard deviation of three individual experiments. ** and *** represent significant difference from control group (** *p* < 0.01; *** *p* < 0.001).

**Figure 8 biomolecules-13-01785-f008:**
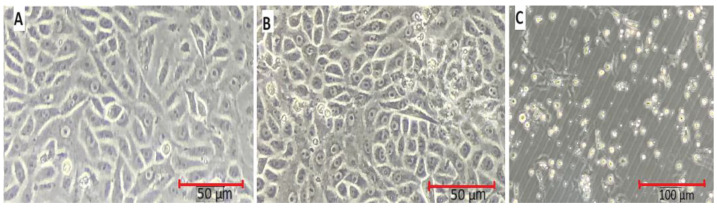
AuNPs induced cellular morphology change in cell lines. *A549* cell lines after treatment in the presence and absence of AuNPs. (**A**) Control. (**B**) Treatment with 5 µg/mL of AuNPs. (**C**) Treatment with 25 µg/mL of AuNPs.

**Figure 9 biomolecules-13-01785-f009:**
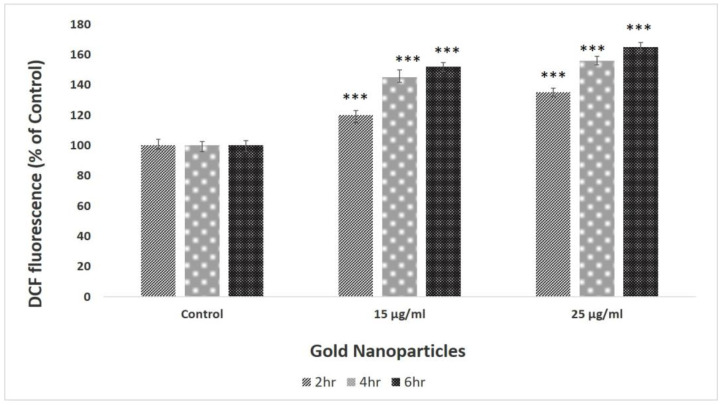
ROS estimation in *A549* cell lines after incubation with AuNPs at various time intervals (2 h, 4 h, 6 h). Data represent the mean ± standard deviation of three individual experiments *** *p* < 0.001 represent significant difference from control group.

**Figure 10 biomolecules-13-01785-f010:**
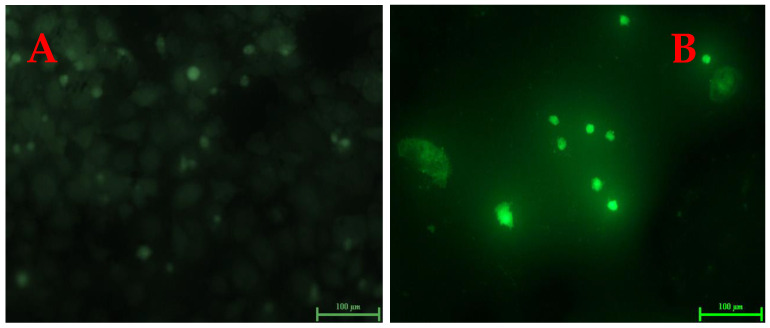
Effect of gold nanoparticles on cell morphology of *A549* cells. The images were taken of DCFDA-stained cells. (**A**) Exposed for 4 h to a 0 µg/mL concentration of AuNPs (control). (**B**) Exposed for 4 h to a 15 µg/mL concentration of AuNPs.

## Data Availability

All data generated or analyzed during this study are included in the published article.

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
