# Peer review of "Biological Synthesis, Characterization, and Therapeutic Potential of S. commune-Mediated Gold Nanoparticles"

_biomolecules, 2023, doi:10.3390/biom13121785_

Round 1
Reviewer 1 Report
Comments and Suggestions for Authors
The current manuscript discussed AuNPs prepared using S. commune and their potential applications as potential anti-fungi and anti-cancer agents.
Below are some of my questions:
1) What has been done in biologically synthesized nanoparticle research? What has been done in biologically synthesized AuNP research? The introduction can benefit from some introductions of related research done.
2) Fig 4: how was the AuNP concentration calculated to have unit in uM?
3) Fig 8 scale bar is missing
4) In the introduction, please include information on what has been done in biologically synthesized nanoparticles, and also biologically synthesized AuNPs.
5) Also discuss why biologically synthesise AuNPs are less cytotoxic to cells than chemically prepared? In discussing this, please also take into consideration that the sizes of these particles would also likely be different.
Comments on the Quality of English LanguageMinor editing of English language required
Reviewer 2 Report
Comments and Suggestions for Authors
Since the authors have revised the manuscript according the previous reviewers' comments, I am pleased to recommend its publicaiton in the journal of biomolecules.
Author Response
I extend my sincere thanks for recommending my manuscript. Your invaluable insights greatly enhanced its quality. Your dedication to the review process is truly appreciated. I am grateful for your time and expertise.
Reviewer 3 Report
Comments and Suggestions for Authors
The research article titled "Biological synthesis, characterization, and therapeutic potential of S. commune-mediated gold nanoparticles" submitted by the authors represents an extension of their ongoing work in nanoparticle synthesis through biological methods. In this comprehensive study, the authors detail the biosynthesis of gold nanoparticles using S. commune, employing a green synthesis approach. The newly synthesized AuNPs are explored for their potential therapeutic applications in treating fungal infections and as potential cytotoxic agents for cancer.
While the introduction is well-crafted with ample references, there is room for streamlining to enhance conciseness. Although the research problem may not be groundbreaking, the study is meticulously designed and executed, supported by robust experimental evidence and data. The synthesis and characterization of AuNPs are particularly well-executed, and the subsequent examination for antifungal activity reveals promising results, albeit at a concentration of 200 μM. It is noteworthy to consider whether this concentration is optimal for novel therapeutic approaches.
The authors further conduct an insightful anticancer activity profile in the A549 cell line, demonstrating promising cytotoxicity. An important consideration is whether the observed antifungal activity is solely attributed to cytotoxicity, and providing clarification on this point would enhance the manuscript.
Overall, the article presents a novel approach to AuNP preparation through a biosynthetic pathway involving S. commune. The systematic explanation of the methodology and the preliminary biological evaluations for antifungal and anticancer activities are commendable. I strongly recommend the publication of this work in Biomolecules.
Author Response

(The authors gave the same response as above.)

Reviewer 4 Report
Comments and Suggestions for Authors
In the manuscript submitted for review entitled "Biological synthesis, characterization
and therapeutic potential of S. commune-mediated gold nanoparticles" The authors used gold
nanoparticles in the study, which they obtained using an extracellular extract from the fungus
Schizophyllum commune. They further examined the obtained nanoparticles using
transmission electron microscopy, dynamic light scattering, and UV-visible spectroscopy.
The authors also assessed the antifungal and anticancer effectiveness of gold nanoparticles.
They showed that they have strong antifungal activity against some pathogenic fungal strains.
Moreover, the obtained gold nanoparticles showed dose-dependent activity against human basal
alveolar epithelial cells with adenocarcinoma (A549), which proves their high biological impact on cancer cells. Congratulations to the authors of a very interesting article! I actually have no objections to this manuscript. However, I noticed a certain irregularity regarding the wording,
among others, in the abstract: "After treating these cancer..." suggests that the cancer cells were treated.
In my opinion, it is safer to write that after 4 hours of incubation with nanoparticles, a significant increase
in intracellular ROS was observed in cancer cells.
I wish you further success.
Round 2
Reviewer 1 Report
Comments and Suggestions for Authors
The method to calculate the AuNP concentration is questionable. The method is assuming that 1 Au atom in solution will form 1 AuNP, which is very unlikely. the authors should change the concentration to mass concentration or quote a reference for their calculation to support their approach.
